# Enhancing Physical and Cognitive Performance in Youth Football: The Role of Specific Dual-Task Training

**DOI:** 10.3390/jfmk10040404

**Published:** 2025-10-18

**Authors:** Juan Miguel Ramírez Lucas, Juan Antonio Párraga Montilla, José Carlos Cabrera Linares, Pedro Ángel Latorre Román

**Affiliations:** Department of Musical, Plastic and Corporal Expression, University of Jaén, 23071 Jaén, Spain; jmrl0014@red.ujaen.es (J.M.R.L.); jccabrer@ujaen.es (J.C.C.L.); platorre@ujaen.es (P.Á.L.R.)

**Keywords:** U16 football players, dual task training, performance, visual attention

## Abstract

**Background:** Football performance depends on the integration of physical, technical, and cognitive abilities under constantly changing conditions. In this context, dual-task training combining physical and cognitive demands has emerged as a promising approach to enhance decision-making and game intelligence in youth football players. **Objective:** The aim of this study was to determine the effects of an eight-week dual-task training programme on physical (speed, strength, and agility), cognitive (working memory, planning, processing speed, and response time), technical (dribbling and short passing), and dual-task performance in U16 football players. **Methods:** Thirty-two players (age: 14.88 ± 0.65 years; BMI: 20.98 ± 1.79 kg/m^2^) were randomly assigned to a control group (n = 14) and an experimental group (n = 18). The experimental group completed a dual cognitive–motor training (CMT) programme consisting of 24 sessions (3 sessions/week, 10–15 min each), integrated into regular football practice. Pre-intervention and post-intervention assessments included football skills (dribbling and passing tests), cognitive tests (Wom-Rest and Vismem-Plan), physical tests (countermovement jump, 20 m sprint, and 505 change-of-direction), and a dual-task test (soccer skills and cognitive aptitude test). **Results:** The experimental group showed significant improvements in all assessed variables, while the control group exhibited no changes or declines in performance. The most notable effects were observed in SoSCAT with visual interference, dual-task cost, and 505 change-of-direction. **Conclusions:** The findings suggest that integrating brief dual CMT programmes into regular football practice can simultaneously enhance physical, technical, and cognitive performance in youth players. This evidence supports the implementation of dual CMT as an effective and time-efficient tool in talent development programmes.

## 1. Introduction

Football is a highly complex sport that requires the integration of physical, technical, and cognitive skills. In open-skill sports such as football, players must make decisions quickly and accurately using optimal skills in a dynamically changing environment [1]. In this context, the ability to efficiently manage dual-tasking (DT) (i.e., the simultaneous execution of a physical and a cognitive task) is critical for optimising sporting performance [2,3].

The scientific literature has shown that training programmes that combine physical and cognitive tasks, known as cognitive–motor training (CMT), have a positive impact on both the motor skills and the cognitive abilities of athletes [4,5]. Therefore, performing DT exercises that resemble various realistic football game situations together with cognitive tasks can help players accumulate experience that can be automatically applied during football matches [2]. In this regard, increased processing speed and accuracy in decision-making can be critical for football players, especially in highly competitive situations where quick and correct decisions can define the outcome of the match [6,7].

The literature suggests that DT has a differential impact depending on the type of cognitive task that is combined with the physical activity. In this sense, Plummer and Eskes [8] conclude that attention is limited when two tasks have to be performed together; consequently, one, or even both, of the tasks slows down. These changes depend not only on cognitive reserve, but also on adequate motor development [9], and this is called interference or DT cost (DTC) [10]. Indeed, CMT can be classified, according to exercise dynamics, into sequential CMT and dual CMT [11]. In sequential CMT, participants perform motor and cognitive exercises successively on the same day or on different days, whereas in the case of dual CMT, athletes perform the two types of exercise simultaneously [11,12]. In athletes, it has been established that dual CMT is more beneficial for improving executive functions than sequential CMT as well as single physical training [4,5]. In addition, there are two types of dual CMT: CMT with additional cognitive tasks and CMT with incorporated cognitive tasks. CMT with additional cognitive tasks is similar to ‘classical’ DT approaches, in which the secondary cognitive task is typically used as a distractor to the motor task [13]. For example, the athlete could be asked to pass the ball to their partner while performing an arithmetic task, which can be described as thinking while moving. In contrast, in CMT with incorporated cognitive tasks, the cognitive task is ‘incorporated’ into the motor task. For example, the athlete could play a small-sided game (8 vs. 2) where the goal is to maintain possession of the ball; however, depending on which player the ball comes from (colours red/yellow = 2 touches, green/blue = 1 touch, player without bib = free touch), the player has to give one touch, two touches or a free touch, which can be described as moving while thinking [11].

Previous studies have applied CMT programmes [14,15]. In a study with a duration of five weeks with semi-professional basketball players, the experimental group showed a significant improvement in their performance compared to the control group, which only performed motor exercises [15]. Similarly, Casella et al. [14] conducted a study with twenty-four football players and concluded that CMT is more effective than conventional motor training, as it improves both planning and visual search abilities. Furthermore, Bherer et al. [16] suggest that this type of training favours the development of new perceptual strategies, optimising attentional focus on task-relevant cues, which in turn contributes to more efficient decision-making. These findings are consistent with the idea that the brain acts like a “muscle”, improving its functional capacity as it faces greater challenges [17,18].

Importantly, performance in football depends on players’ abilities to continuously perceive and act on information within a dynamic and time-limited environment [19]. Furthermore, players’ ability to share the focus of attention between different cues in the environment (the ball, opponents, and teammates) is essential for successful decision-making [20,21]. However, the current scientific literature remains limited concerning the effects of dual CMT on both physical and cognitive performance in young soccer players [2,22]. The study conducted by Casella et al. [14] compared the effect of CMT and a conventional training programme on planning and visual search skills over ten weeks in 24 ten-year-old football players. The study reported that CMT was more effective than motor training alone in improving the cognitive functions of the players. Similarly, the research carried out by Baccouch et al. [2] compared the effect of dual CMT and a conventional training programme on change-of-direction skills (CODA) and cognitive performance (cognitive flexibility and inhibition) over four weeks in 24 13-year-old football players. After training, CODA and cognitive flexibility and inhibition improved only in the group performing dual CMT. These results are especially relevant in contexts where players are under significant stress and must maintain high levels of both physical and cognitive performance.

Recently, the scientific literature has expanded this line of research by exploring the DT paradigm in youth football players, integrating cognitive, technical, and physical variables. Esposito et al. [22] demonstrated that psychokinetic training, combining technical drills with perceptual–cognitive demands, enhanced coordination and attentional control in youth footballers. In addition, Latorre et al. [3] validated a soccer-specific DT assessment (SoSCAT) that integrates technical execution with cognitive processing in players aged 12–18 years. Likewise, Fiebre et al. [23] confirmed strong relationships between agility and decision-making in elite youth players. Studies have also analysed the acute influence of DT constraints during small-sided games, showing alterations in tactical behaviour and decision-making quality [24]. Similarly, Klotzier and Scott [25] conducted a study with two hundred and seventy-five participants, concluding that DT dribbling tests can discriminate between performance levels, especially in soccer players aged 14–17 years old.

In addition, technical skills such as dribbling and short passing are crucial determinants of football performance at youth levels, as they require the continuous integration of perceptual cues, motor control, and rapid decision-making under time constraints [3]. These skills demand the coordination of visual attention, motor execution, and tactical awareness, reflecting the essence of DT performance [24]. Hence, training approaches that combine technical and cognitive components have been shown to enhance both technical precision and cognitive efficiency in young footballers [22,23]. Notice that the current study was conducted with semi-professional youth football (at least six years of federated experience) academy competing in the U16 regional league. Players trained three times/week (90 min/session) plus one official match at the weekend (the club’s training methodology is described in greater detail in Section 2.4).

Furthermore, physical fitness plays a decisive role in youth football performance, particularly in relation to sprinting, agility, jumping, and change-of-direction ability. Previous research has shown that these attributes develop progressively during adolescence and are closely associated with biological maturation and technical proficiency [26,27]. Aloui et al. [27] concluded in their research that combining plyometric and short sprint training in U15 players produced significant improvements in sprinting, jumping, and change-of-direction performance. In line with these findings, the recent systematic review by [28] confirmed that adolescent football players’ experience increases in jumping, sprinting, and agility capacities following the implementation of plyometric training programmes. Furthermore, it has been reported that during a match, most football actions last less than five seconds and rarely exceed ten seconds [29], which reinforces the importance of developing these explosive capabilities at these ages. Additionally, Clarke et al. [30] concluded that linear sprint speed shows the strongest correlation with performance in the 505 test, a recognised measure of change-of-direction ability. Collectively, these findings support the inclusion of speed, strength, and agility variables in the present study, as they represent key components for enhancing performance during this formative stage. In this sense, it is relevant to investigate the effects of dual CMT on the physical–cognitive performance of young football players, who are at a crucial stage of development both physically and mentally [31].

Therefore, the aim of this study was to determine the effects of a DT training programme on physical (speed, strength, and agility), cognitive (working memory, planning, processing speed, and response time), technical (driving and short passing), and performance skills in a DT environment in U16 football players. The underlying hypothesis is that, after eight weeks of training, football players who participate in the combined training programme with DT will show significant improvements in physical, technical, and cognitive performance as well as improvements in the DT compared to players who only undergo conventional football training.

## 2. Materials and Methods

### 2.1. Participants

A total of 32 male players (mean age: 14.88 ± 0.65 years old; height: 1.74 ± 0.06 m; BMI: 20.98 ± 1.79 kg/m^2^; experience as federated football players: 7.48 ± 1.81 years) agreed to participate in this study. The positions of the players were goalkeepers: 2; defense: 8; midfielder: 10; and forward: 12. The players had to participate in at least 80% of the training sessions during the eight weeks of the intervention to be included in the analysis. Inclusion criteria were as follows: (1) to have a national football license; and (2) not to have suffered injuries affecting football performance in the three months prior to testing. Exclusion criteria for the study were as follows: (1) participation in any additional training programme during the intervention weeks; and (2) any illness or injury that prevented completion of the study protocol. Parents voluntarily signed an informed consent form for their children’s participation in this study. The study was conducted in accordance with the guidelines of the Declaration of Helsinki [32] and was approved by the Ethics Committee of the University of Jaén (Ref: 20231218/ENE.TES).

This study was conducted during the regular competitive season and adopted a randomised longitudinal experimental design with pre-test and post-test measures, including a control group (CG) and an experimental group (EG), in line with recent interventions carried out in youth football players [2,33]. The intervention lasted eight weeks, during which the EG completed three weekly DT training sessions (10–15 min per session) integrated into their regular football practice, while the CG continued with their standard technical–tactical training.

Random allocation ensured baseline comparability between groups in terms of age, anthropometric characteristics, and playing experience. This design was chosen to examine the effects of DT training on physical, cognitive, and technical performance variables in U16 players.

Differences in the final number of participants between groups were due to the exclusion of players who did not meet the inclusion criteria (n = 1), personal reasons (n = 1), injuries sustained during the intervention (EG = 2; CG = 3), and absence from the post-test session (CG = 2). Ultimately, a total of 18 players in the EG and 14 in the CG completed the study (Figure 1).

### 2.2. Material and Testing

#### 2.2.1. Anthropometric and Sociodemographic Measures

Body mass was measured using a weighing scale (Seca 899, Hamburg, Germany), and body height was measured with a stadiometer (Seca 222, Hamburg, Germany). The participants’ body mass index (BMI) was calculated by dividing their body mass (in kilograms) by the square of their body height (in metres). Football-related data were collected, including experience in federated football (in years), category, number of goals scored during the season (in league matches), and usual position on the pitch.

#### 2.2.2. Football Skill Test

Football abilities were measured using a dribbling and a passing test The results of the dribbling and passing tests were combined to give the variable ∑skill (passing + dribbling) [26]. Both tests were performed with official balls, and test times were obtained using a Witty photocell. In the dribbling test (Figure 2), a participant with a ball ran straight to a post which was located 20 m from the starting line. The player had to have at least three touches before they reached the post. Subsequently, the participant dribbled back to the starting line between poles which were placed at 4 m, 2 m, 2 m, 4 m, 2 m, 2 m, and 4 m from each other. The same straight-run dribbling action was then repeated from the other side to complete the test. One successful attempt was required, and the best of two attempts was selected for analysis unless the player failed to complete the test within 50 s, in which case 50 s was registered as the performance time. The test/re-test reliability of the dribbling test was 29.6 ± 3.2 s vs. 30.0 ± 3.1 s with a correlation coefficient of r = 0.82 (*p* < 0.001) [26].

The passing test (Figure 3) was performed with two cones that were 6 m apart and passing walls with width 2 m placed 7 m away from the cones. The test started when the participant kicked the first pass against the wall, then received the rebound, and dribbled between the cones, then immediately repeated the cycle. The test ended when the tenth pass hit the passing wall. Five of the passes had to be made with the right foot and five with the left foot. One successful attempt was required in the test, and the best out of two attempts was selected for analysis unless the player failed to complete the test within 60 s, in which case 60 s was registered as the result. The test/re-test reliability of the passing test was 45.0 ± 6.7 s vs. 45.4 ± 6.7 s, r = 0.81 (*p* < 0.001) [26].

#### 2.2.3. Cognitive Test

To analyse cognitive function, the computerised Vismem-Plan test was used, which is based on the cube test [34]. In this test, the participant must remember the greatest sequence of visual stimuli possible. A total of ten stimuli are shown arranged irregularly across the screen. In addition, the Wom-Rest recognition test was used, which is based on the classic symbol-search tests (WAIS). In this test, the participant is required to memorise the stimuli that are presented and then recognise which were the original stimuli among the distractors. Both tests measure spatial perception, short-term memory, visual short-term memory, non-verbal memory, working memory, planning, processing speed, and response time, among other abilities. Both tests were applied using the following web tool: https://www.cognifit.com/es/bateria-de-pruebas-y-tareas/test-vismem-plan/test-de-concentracion (access on 19 February 2025).

#### 2.2.4. Physical Test

##### Countermovement Jump (CMJ)

Jump height was calculated from flight time using an infrared platform (Optogait, Microgate, Bolzano, Italy) that estimates the jump height from the flight time [35]. During the CMJ, the subject was instructed to rest his hands on his hips while performing a downward movement to reach about 90° of knee flexion, followed by a maximal vertical jump [36].

##### Twenty-Metre Sprint Test

The running speed was calculated from the time (in seconds) measured using two double light barriers (WITTY photocell; Microgate Srl, Bolzano, Italy; accuracy 0.001 s), which were placed at the beginning and the end of a corridor [37]

##### The 505 Change-of-Direction Test (505 COD)

The test was performed by sprinting for 10 m and then pivoting and sprinting back to the starting line [38]. Time was measured using a WITTY photocell. The average for the 505 COD was calculated from the following formula: 505 COD average = Time 505 COD with right leg + Time 505 COD with left leg [30].

#### 2.2.5. Dual Test

##### Soccer Skills and Cognitive Aptitude Test (SoSCAT)

The SoSCAT has been previously validated by Latorre-Roman et al. [3], showing adequate reliability and validity parameters. The test was conducted in two different conditions (i.e., with and without visual interference).

The test starts with a 10 m sprint, and then a slalom between four cones (jinking run zone) had to be performed. Subsequently, the participants had to turn right or left (in this condition, the participant could choose any direction to go), turn around the cone (change-of-direction zone) and come back to the driving line. After that, they had to continue to run forward until reaching a kick area where they had to kick the ball against a wooden wall (they could decide right or left), simulating a partner’s pass. Once the ball kicked the wall and it was received by the player, the participant had to reach the last cone and turn around. Then, they had to retrace the same track until reaching the finish point, and then they had to kick the ball (launch area) between the space created by three cones (right or left), representing a goal simulation.

To apply visual inference, five WITTY photocells and five WITTY SEM lights were included in the critical areas of the test, such as the change-of-direction area, kicking area, and shooting area. The traffic lights were activated each time the player crossed the witty photocell, allowing the player to watch the instruction properly, since the traffic lights were programmed to appear with two different options (red or green arrow). They had to follow the direction indicated by the arrow when the traffic light showed the green arrow. However, if the traffic light showed the red arrow, they had to go in the direction opposite to that indicated by the arrow (Figure 4). The SoSCAT was performed with official balls, and some familiarisation testing was performed without interference.

### 2.3. Procedure

All the football performance tests were assessed before and after the dual CMT intervention, and all testing sessions of the study were conducted between 19:00 and 21:00 h to ensure comparable environmental and physiological conditions across participants and to coincide with their regular training schedule. The football players were tested during the competitive season. Testing sessions were scheduled on Tuesday, Wednesday, and Friday, avoiding match days and ensuring at least 48 h of recovery between tests. No vigorous physical activity was performed on the day before testing.

All tests were carried out in the participants’ usual training environment (i.e., on an artificial grass football pitch). Each DT training session lasted approximately 10–15 min and was integrated into the regular 90 min football training sessions. Three separate days were used to complete the entire evaluation protocol.

On the first day, the sociodemographic questionnaire was administered, and body weight and height were recorded. On the second day, after a standardised 10 min warm-up—which included light aerobic running, changes of direction, and progressive sprints—participants received instructions, and the research team presented the study protocol. Two familiarisation attempts were allowed for all tests conducted. No feedback was provided to the participants between recorded attempts. The order of the tests after the warm-up was as follows: (1) CMJ; (2) 20 m sprint test; (3) 505 COD; (4) football dribbling test; (5) football passing test; and (6) SoSCAT test. In the SoSCAT, interference was randomised so that some players started with visual interference and finished without it, while others performed the test in the reverse order. For the analysis, the best of each player’s attempts was selected. In all trials, players rested for five minutes between each attempt to ensure full recovery. On the third day, the cognitive tests (Vismem-Plan and Wom-Rest) were administered individually in a quiet room that allowed players to concentrate properly.

### 2.4. Intervention

The DT programme was conducted at the beginning of each weekly session. EG performed 24 DT sessions (3 days/week × 8 weeks) for a duration of 10–15 min, whereas the CG continued performing their training program (i.e., running at different intensities, dribbling…). The usual training during the intervention consisted of a small-sided game, followed by medium possession (40 × 40 m) and large-space possessions (half football pitch). The sessions usually ended with attack–defence games or modified matches. All tasks were aimed at stimulating executive functions through motor–cognitive exercises. The EG were asked to perform different DT exercises, which required (I) various football-specific skills such as agility, sprinting, acceleration, deceleration, accuracy, passing, driving, ball control, ball keeping, ball preservation, offensive/defensive transitions, tilting, permanent help, covering, tackling, anticipation, and interception; and (II) various cognitive functions such as attention, concentration, perception of external cues, cognitive flexibility, inhibition, working memory, and decision-making. To promote these cognitive functions, the research team used different tools (cones, training bibs, headbands, and eye patches) of different colours and emitted different auditory signals (with various loudspeakers along with an audio recording).

The first day of the training week consisted of a series of small-sided games or ball possession tasks that gradually incorporated elements of cognitive and motor difficulty. Different variations were introduced, such as colour identification, number sequences, mathematical operations and auditory cues, which required players to make quick decisions. The second day of weekly training focused on passing wheels with cognitive elements, integrating arithmetic tasks to develop attention and dual-processing skills. These exercises started with simple calculations, such as counting backwards by two or three numbers, and became progressively more complex in subsequent weeks including addition, subtraction, multiplication, and division operations in different sequences. On the last training day of the week, the focus was on speed drills with cognitive games or one in the middle in a DT context, where players had to react to complex instructions given by the coach, such as changes of direction, colour and number stimuli, and even solving simple mathematical problems while trying to escape from an opponent.

All these DT and their progressions allowed players to adapt to game situations with a high cognitive load environment, which, in addition to improving their technical skills, stimulated their responsiveness in game contexts. The participants were verbally instructed to maintain attention on the motor and cognitive tasks simultaneously and never to stop a motor exercise to perform a cognitive task. The EG group were given the opportunity in a pre-training session to learn the correct procedure to perform the DT situations. The DT situations were usually two sets over five minutes with a rest of one or two minutes depending on the exercise difficulty (Table 1).

The exercises in the DT programme were not individualised, as the programme was designed for group sports training, in this case football. Moreover, the tasks followed a teaching progression from less to more difficult. During each training session, the coach verbally and positively encouraged the participants to induce their maximum effort. The participants were considered to have completed the DT training programme if they participated in at least 80% of the training sessions.

### 2.5. Statistical Analysis

The data were analysed using SPSS, v. 22.0 for Windows (SPSS Inc., Chicago, IL, USA), and the significance level was set at α = 95%. The data are shown as descriptive statistics, including mean and standard deviation (SD). Tests of normal distribution and homogeneity (Kolmogorov–Smirnov and Levene’s, respectively) were conducted on all data before analysis. A 2 × 2 analysis of variance (ANOVA) with repeated measures (group × measurement) was conducted for the dependent variables. Additionally, effect sizes for group differences were expressed as Cohen’s *d* and are reported as trivial (<0.2), small (0.2–0.49), medium (0.5–0.79), and large (≥0.8) [39]. Pearson’s correlation analysis was conducted between the changes (∆, e.g., DTC at post-test—DTC at pre-test) found for all variables.

## 3. Results

Table 2 presents a comparison between the anthropometric and sociodemographic characteristics of the two groups analysed. No statistically significant differences were found between the two groups in any of the variables.

Table 3, which analyses the performance in football skills and cognitive measures of the players in the CG and EG, shows significant differences before and after the intervention in several tests. The 2 × 2 ANOVA revealed significant effects for the group × time (group comparison: CG vs. EG). Both groups showed similar values at pre-test (*p* ≥ 0.05), except that in SoSCAT with visual interference and 505 COD, the EG showed a significantly better performance than the CG. In post-test, the EG displayed higher scores than the CG except for CMJ, the passing test, and Vismem-Plan—for these variables, no significant differences appear.

Regarding the time × group interaction (within-group), the EG experienced significant improvements in all the variables analysed. However, the CG did not experience significant changes in most variables, and they showed worse performance in SoSCAT with visual interference, DTC, and 505 COD.

In relation to the post-test–pre-test differences (increase), the EG showed greater improvements than the CG in most variables except SoSCAT without interference, the passing test, and ∑skill (= passing + dribbling), where no significant differences were found.

The correlation analysis reveals various significant associations between ΔDTC and Δ505 COD average (r = 0.469, *p* = 0.007), ΔWom-Rest (r = −0.436, *p* = 0.013).

## 4. Discussion

The aim of this study was to determine the effects of an eight-week DT training programme on physical (speed, strength, and agility), cognitive (working memory, planning, processing speed, and response time), technical (dribbling and short passing), and DT performance in U16 football players. The findings in the present study confirmed that the EG, after eight weeks of dual CMT, showed significant improvements in the physical, technical, cognitive, and dual environment performance tests, while the CG experienced no such improvements and even performed worse in some skills. These results align with previous research highlighting the benefits of DT training on both technical performance and cognitive integration under pressure [5,14].

The observed improvement in DTC for the EG and the worsening in the CG reinforces the idea that dual training is essential for improving the ability to manage multiple simultaneous demands [15,25]. The ability of players to maintain the quality of their technical skills while performing cognitive tasks under pressure is crucial for performance in football, where game situations demand rapid physical and cognitive responses [1,22]. Although recent studies indicate that superior working memory capacity does not invariably lead to better performance in DT [40], most studies confirm the importance of working memory capacity in improving DT performance [40] Although our findings on working memory improvement align with those reported by Casella et al. [14] and Baccouch et al. [2], who observed similar gains in executive functions after DT interventions, future research should directly examine the specific contribution of working memory to overall DT performance in youth footballers. In this sense, these results are in agreement with these investigations, as the footballers who participated in the current study showed significant improvements in their performance under DT conditions. However, further studies are needed to fully unravel the mechanisms underlying this result.

As for the physical tests, the results of the 20-metre sprint test and the 505 change-of-direction test showed that the EG experienced significant improvements in their speed and ability to change direction in an agile manner, while the CG did not show significant progress. Specifically, the EG showed a significant improvement of 0.09 s (*p* = 0.001) in the 20 m sprint test, and this supports the idea that programmes that integrate cognitive tasks with physical training can not only maintain but also improve physical performance. These results are in concordance with the study conducted by Baccouch et al. [2], since after a DT training programme, the execution time for an agility test showed a significant decrease (*p* < 0.001) for the EG, who underwent dual CMT, but not for the CG (*p* = 0.38), who performed conventional football training. In the CMJ test, the EG improved their jumping power, while the CG remained stable.

Regarding technical tests, both groups improved in ball handling and the passing test, although the EG obtained better results. The improvement observed in dribbling and short passing accuracy in the EG is in line with the results of Esposito et al. [22], who found that psychokinetic training combining technical and perceptual–cognitive components enhanced coordination and attentional control in youth footballers. These findings reinforce the idea that DT contexts foster greater adaptability and technical precision under pressure. This more pronounced improvement in the EG indicates that the incorporation of cognitive tasks during training favours not only physical development but also technical improvement under conditions of greater complexity, as suggested by recent studies showing that dual CMT enhances sport-specific skills [41].

In relation to the cognitive tests, the EG showed improvements in the Wom-Rest, while the CG experienced a decrease in performance; however, no significant differences were found for the Vismem-Plan. Although improvement was only observed in one cognitive test, dual CMT training could enhance skills such as planning, sustained and divided attention, inhibition, and working memory capacity. These skills are important for the performance of a football player, who must constantly assess the situation, compare it with previous experiences, create new possibilities, and make quick decisions to act, as well as quickly inhibit planned decisions [42]. In this regard, previous studies have indicated that training based on the integration of cognitive demands can enhance executive function in young athletes, improving their ability to process information under stress and make effective decisions [5,22]. In agreement with our results, the study by Casella et al. [14] showed that CMT applied to football practice was more effective than motor training alone in improving other specific executive functions (planning and visual search skills) in 10-year-old football players. Also, the study by Baccouch et al. [2] showed that dual CMT applied to football practice was more effective than conventional football training in improving cognitive performance (flexibility and cognitive inhibition) in 13-year-old football players. Furthermore, in the research of Fleddermann et al. [43], the improvement in attentional control efficiency after a perceptual–cognitive intervention in volleyball athletes may be supported by other improved cognitive processes, such as the ability to sustain attention and processing speed. The effects observed in our U16 football players after dual CMT are likely to be due to improvements in functions such as working memory, planning, processing speed and response time. This is especially relevant considering that training that integrates both physical and cognitive resources has been shown to be effective in promoting functional neuroplasticity [15].

To sum up, the results of the present study support the idea that dual CMT programmes not onlsy improve physical performance but also strengthen the cognitive and technical abilities necessary for success in football [14]. Taken together, these findings demonstrate that combining physical and cognitive training (24 sessions divided into 3 days/week × 8 weeks) within realistic football contexts can accelerate both neuromotor and decision-making development in youth players, supporting the long-term goal of enhancing game intelligence and performance efficiency [24]. This reinforces the idea that a comprehensive approach, combining physical and cognitive demands, is crucial for the development of young football players because the development of executive functions takes place progressively throughout childhood and adolescence from birth to 19 years of age [44]. As the scientific evidence argues, dual CMT emerges as a powerful complementary training tool within the sphere of sports training and can improve cognitive–motor performance in players participating in cognitively demanding sports such as football [2].

### Limitations and Strength

In the current study, there are some limitations that should be mentioned: (1) the study was conducted with a limited number of participants, which could affect the generalisability of the results. Future research should include larger and more diverse samples to obtain more representative results; (2) all the study participants were male, which limits the applicability of the findings to female football players. Differences in physical and cognitive development between males and females could influence how dual CMT impacts female football players; (3) although the study demonstrated significant improvements after eight weeks of training, no long-term follow-up was conducted to determine the persistence of these effects. Longitudinal studies covering one or more competitive seasons would help to establish whether dual cognitive–motor training produces lasting benefits or requires continuous implementation.

Additionally, this study did not account for the potential influence of playing position, which may affect both the physical and cognitive demands experienced during training and testing. Future investigations should consider position-specific responses to DT training, as tactical roles and perceptual–motor requirements vary across defenders, midfielders, and forwards. Moreover, future research should aim to replicate these findings using randomised controlled trial designs to strengthen causal inference.

Despite these limitations, a strength of the present study lies in its comprehensive assessment of football performance, integrating physical, technical, cognitive, and DT variables within a single intervention. This multidimensional approach provides valuable evidence on how DT training can optimise performance in sports with high cognitive–motor demands such as football. From a practical standpoint, the significant improvements observed in the EG across all domains suggest that this type of programme can be effectively incorporated into regular youth football training routines.

## 5. Conclusions

In conclusion, this study shows that an eight-week dual cognitive–motor training programme (24 sessions divided into 3 days/week × 8 weeks) in U16 football players produces significant improvements in physical, technical, and cognitive performance, especially in DT conditions. These improvements optimise the players’ physical and technical ability, as well as their decision-making ability under pressure.

Beyond demonstrating performance gains after 24 training sessions, the present findings highlight the importance of integrating cognitive–motor demands into regular football practice to foster adaptable, intelligent, and resilient players. This approach not only enhances technical precision and perceptual–cognitive efficiency but also contributes to more effective decision-making in dynamic, high-pressure situations.

From a practical standpoint, incorporating short DT drills within standard training sessions may represent an efficient and evidence-based strategy to improve multiple performance domains simultaneously. However, future research should explore the durability of these effects through long-term evaluations, examine differences according to playing position and competitive level and extend this approach to female players.

### Practical Applications

From a practical perspective, the implementation of these training programmes in sports clubs would offer a valuable tool for coaches, physical trainers, and psychologists by facilitating the integral development of players. This would not only optimise athletic performance in high-level football but also foster a holistic approach from an early age, enhancing both the physical and the cognitive skills of young football players at key stages of their development.

## Figures and Tables

**Figure 1 jfmk-10-00404-f001:**
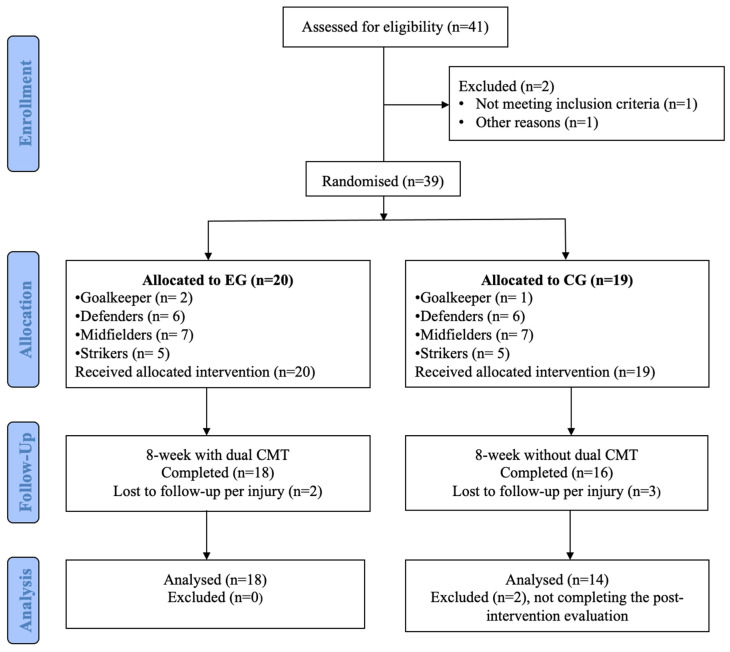
Study design flowchart.

**Figure 2 jfmk-10-00404-f002:**
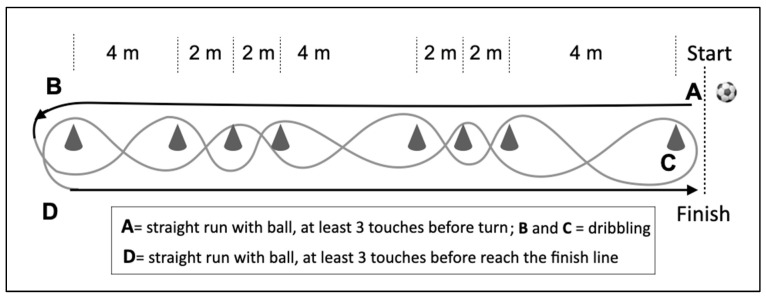
Dribbling test.

**Figure 3 jfmk-10-00404-f003:**
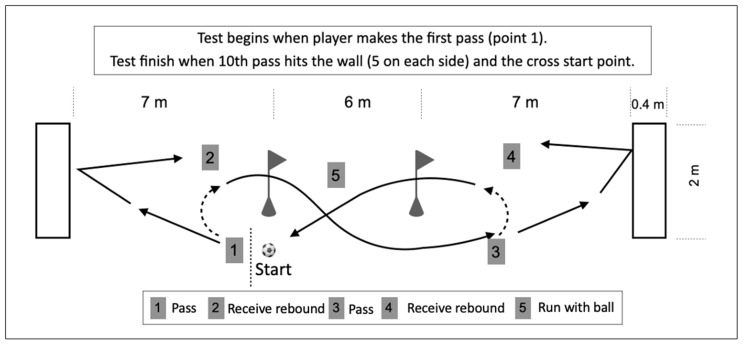
Passing test.

**Figure 4 jfmk-10-00404-f004:**
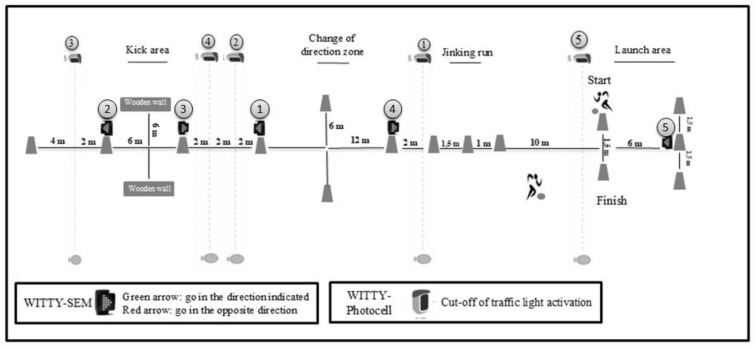
Soccer skills and cognitive aptitude test. Numbers 1–5 indicate the Witty SEM devices the player had to watch to perform each action. Each device was linked to a photocell (1–5) that triggered the command (green or red arrow) when the player crossed it.

**Table 1 jfmk-10-00404-t001:** Dual cognitive–motor training programme.

Football-Specific Dual Training Programme							
Week 1	Week 2	Week 3	Week 4	Week 5	Week 6	Week 7	Week 8
First day of the week (dual small-sided games and dual ball possessions)							
Day 1	Day 4	Day 7	Day 10	Day 13	Day 16	Day 19	Day 22
SSG 8 × 2 With bib Time: 2 sets of 5′ + 1′R	SSG 8 × 2 With headband Time: 2 sets of 5′ + 1′R	SSG 8 × 2 Auditory signals Time: 2 sets of 5′ + 1′R	SSG 8 × 2 Repeat sequence Time: 2 sets of 5′ + 1′R	Ball possession 4 × 4 + 2 UP With headband Time: 2 sets of 4′ + 1′R	SSG 8 × 2 Counting Time: 2 sets of 5′ + 1′R	SSG 8 × 2 Simple operations Time: 2 sets of 5′ + 1′R	Ball possession 4 × 4 + 2 UP Without bibs Time: 2 sets of 4′ + 1′R
Second day of the week (Dual technical figures)							
Day 2	Day 5	Day 8	Day 11	Day 14	Day 17	Day 20	Day 23
Technical figure ‘square’. Countdown Time: 2 sets of 5′ + 1′R	Technical figure ‘triangle’ Countdown and forward Time: 2 sets of 5′ + 1′R	Technical figure ‘Y’ Add Time: 2 sets of 5′ + 1′R	Technical figure ‘Y’ Subtract Time: 2 sets of 5′ + 1′R	Technical figure ‘Y’ Arithmetic operations Time: 2 sets of 5′ + 1R’	Technical figure ‘Y’ Arithmetic operations Time: 2 sets of 5′ + 1R’	Technical figure ‘rhombus’ Arithmetic operations Time: 2 sets of 5′ + 1R’	Technical figure ‘rectangle’ Arithmetic operations Time: 2 sets of 5′ + 1R’
Third day of the week (Dual speed games and dual one in the middle)							
Day 3	Day 6	Day 9	Day 12	Day 15	Day 18	Day 21	Day 24
Reaction speed game Mushroom game Time: 10–15′	One in the middle 4 × 1 with sprints Time: 10–15′	Reaction speed game 2 Odd and even number game. Time: 10–15′	One in the middle 4 × 1 with one eye patches. Time: 10–15′	Reaction speed game 3 directions and colours game “1”. Time: 10–15′	One in the middle 4 × 1 with one eye patches and bibs. Time: 10–15′	Reaction speed game 4 chase and escape game. Time: 10–15′	Reaction speed game 5 directions and colours game “2”. Time: 10–15′

SSG: small-sided game; R: rest; UP: utility players.

**Table 2 jfmk-10-00404-t002:** Anthropometric and sociodemographic characteristics of the participants.

	Total (n = 32) Mean (SD)	CG (n = 14) Mean (SD)	EG (n = 18) Mean (SD)	*p*-Value	Cohen’s D
Age (Years)	14.88 (0.65)	15.10 (0.61)	14.67 (0.69)	0.094	0.676
Weight (kg)	64.05 (7.82)	64.07 (6.32)	63.94 (9.33)	0.965	0.016
Height (m)	1.74 (0.06)	1.74 (0.07)	1.75 (0.06)	0.524	0.160
BMI (Kg/m^2^)	20.98 (1.79)	21.23 (1.69)	20.73 (1.89)	0.448	0.286
Nº years playing football (before federated)	3.94 (0.82)	4.21 (1.05)	3.67 (0.59)	0.072	0.679
Nº years playing football (federated)	7.48 (1.81)	6.86 (2.35)	8.11 (1.28)	0.063	0.708
Nº of goals in the league matches	1.55 (2.26)	1.21 (1.97)	1.89 (2.56)	0.422	0.302

SD: standard deviation; CG: control group; EG: experimental group; BMI: body mass index; Federated implies playing football and being registered in an official competition.

**Table 3 jfmk-10-00404-t003:** Performance in soccer skills and cognitive measures of players in the control and experimental groups.

Variables	Groups	Pre-Test Mean (SD)	Post-Test Mean (SD)	*p*-Value (Time × Group)	Cohen’s D	(Δ) Post-Test-Pre-Test	*p*-Value
SoSCAT without interference	CG	33.74 (3.25)	33.91 (2.68)	0.685	0.057	0.17 (1.47)	0.092
	EG	31.69 (2.48)	30.92 (2.23)	0.039	0.327	−0.78 (1.56)	
*p*-value (group × time)		0.053	0.002				
SoSCAT with visual interference	CG	39.03 (2.50)	40.06 (2.61)	0.027	0.403	1.03 (1.11)	<0.001
	EG	36.66 (2.44)	34.63 (3.06)	<0.001	0.733	−2.03 (1.53)	
*p*-value (group × time)		0.021	<0.001				
DTC	CG	16.22 (8.83)	18.37 (5.90)	0.027	0.286	2.14 (5.06)	0.011
	EG	15.90 (5.68)	12.04 (6.47)	<0.001	0.634	−3.85 (5.51)	
*p*-value (group × time)		0.784	0.044				
20-metre sprint test	CG	3.16 (0.19)	3.19 (0.21)	0.207	0.149	0.02 (0.09)	0.001
	EG	3.15 (0.10)	3.06 (0.10)	0.010	0.9	−0.08 (0.07)	
*p*-value (group × time)		0.756	0.035				
505 COD (average of both) (s)	CG	2.65 (0.30)	2.69 (0.32)	0.005	0.128	0.03 (0.05)	<0.001
	EG	2.45 (0.12)	2.40 (0.12)	<0.001	0.417	−0.05 (0.03)	
*p*-value (group × time)		0.013	0.001				
CMJ (cm)	CG	32.74 (3.32)	32.05 (3.12)	0.216	0.214	−0.69 (1.38)	<0.001
	EG	30.32 (4.96)	33.26 (4.72)	<0.001	0.607	2.94 (2.44)	
*p*-value (group × time)		0.127	0.414				
football dribbling test (s)	CG	27.84 (2.53)	27.46 (2.09)	0.343	0.164	−0.38 (1.55)	0.022
	EG	27.63 (2.04)	25.97 (1.40)	<0.001	0.949	−1.66 (1.43)	
*p*-value (group × time)		0.793	0.022				
football passing test (s)	CG	44.36 (3.12)	43.13 (3.94)	0.065	0.346	−1.23 (3.01)	0.883
	EG	42.54 (2.57)	41.18 (1.49)	0.023	0.647	−1.36 (1.81)	
*p*-value (group × time)		0.080	0.063				
∑skill (=passing + dribbling)	CG	72.20 (5.13)	70.59 (5.63)	0.059	0.299	−1.61 (3.79)	0.210
	EG	70.17 (3.50)	67.15 (2.26)	<0.001	1.025	−3.02 (2.38)	
*p*-value (group × time)		0.192	0.025				
Wom-Rest (%)	CG	77.14 (15.21)	71.57 (14.56)	0.141	0.374	−5.57 (14.10)	0.002
	EG	76.11 (16.07)	87.28 (12.10)	0.002	0.785	11.17 (13.53)	
*p*-value (group × time)		0.855	0.002				
Vismem-Plan (%)	CG	65.00 (8.77)	61.43 (15.86)	0.242	0.278	−3.57 (15.74)	0.013
	EG	61.78 (14.88)	68.72 (12.34)	0.013	0.508	6.94 (5.61)	
*p*-value (group × time)		0.479	0.154				

SoSCAT: soccer skills and cognitive aptitude test; DTC: dual task cost; 505 COD: 505 change-of-direction test; CMJ: countermovement jump; Wom-Rest: cognitive test; Vismem-Plan: cognitive test.

## Data Availability

The data that support the findings of this study are available from the corresponding author upon reasonable request.

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
