# Peer review of "Enhancing Physical and Cognitive Performance in Youth Football: The Role of Specific Dual-Task Training"

_jfmk, 2025, doi:10.3390/jfmk10040404_

Round 1
Reviewer 1 Report
Comments and Suggestions for Authors
Introduction
Add (DT) the fist time that appears.
Methods
Only one researcher measured the time with a stopwatch? Include this in limitations section.
In particular, I am concerned about the assessment of timing in the technical test with the ball. The authors used the Witty device (traffic light system) for the decision-making test, but instead used a stopwatch to measure execution speed in the other task.
Author Response
Dear reviewer,
We would like to thank you for the time spent reviewing our manuscript. We have considered all suggestions and we believe our manuscript is stronger as a result of the changes that we have introduced in the revised version of the manuscript. Changes to the original manuscript are highlighted in yellow font, also you can find in this document an itemized point-by-point response to your comments.

Reviewer 2 Report
Comments and Suggestions for Authors
First of all, I would like to thank you for invited to read the document.
The authors have done an excellent job. Each of the comments shared are intended to improve the study.
The comments can be found in the PDF document.
Also, some of the comments on some of the points that need to be reworded in the paper are shared below:
Abstract
It is suggested that an introductory section be added to reflect the importance of this study.
It´s suggest reviewing the results and conclusions because the distinction between what is shared is not clear. We also suggest adding the training program (number of weeks, duration, characteristics, etc.) to the methodology.
Introduction
If the study focuses on youth soccer, it is suggested that the possibility of incorporating studies that analyze this age group with the variables analyzed in the present study be reviewed.
The context of the players needs to be detailed, and a few paragraphs analyzing the technical variables should be added.
It is also suggested that studies related to the analysis of physical fitness variables in youth soccer players be added.
Line 30-31. It is suggested that some references to studies that have addressed the dual task be added.
Line 34-35. These types of phrases need some context in order to define what is new.
Line 38-40. It is suggested that studies related to soccer, especially youth soccer, be used. Reference is made to a study on rugby (see reference 7).
Material and method
It is suggested that the design and type of study conducted be mentioned. Additionally, add a reference to support it.
Design
Participants
It´s suggest adding a table describing the main characteristics of the athletes (age, body mass, height, BMI, years of experience, playing positions, etc.).
Line 104-106.
Criteria related to the athletes' experience were taken into consideration, for example, having a minimum of six months' experience playing soccer, etc.
In addition, it was taken into consideration that some athletes will train without competing, so it was decided that only those who will do both would be included.
Line 176-177. How many familiarization tests were run before the test?
Procedure
It is suggested that the concepts of technical tests and soccer tests be reviewed and standardized.
What time were the tests conducted?
On what days were the tests conducted?
How long did each training session last?
Discussion
It is suggested that the entire discussion be reviewed in order to analyze the results in light of scientific advances that have been researched in youth soccer players.
It is also suggested to mention the effects of 24 training sessions.
Line 58-62. The study analyzed some variables related to working memory, because it is difficult to assert that the results are related to other studies that do analyze working memory.
Line 63-72. What was the result of the 20-meter sprint to discuss with other studies in youth soccer?
Line 73-78. It´s suggest reviewing the discussion in this section, as it does not mention the characteristics of the technical variables in this study or in other research.
Limitations and strength
It is suggested that limitations and future prospects be reviewed in order to analyze topics related to future studies that examine physical, technical, and cognitive variables in response to playing position, longitudinal studies over one or more seasons, and randomized controlled trials.
Conclusions
It is suggested that the conclusions be reviewed in order to contribute the relevant findings of the study beyond mentioning that there were differences in 24 training sessions.
References
It´s suggest reviewing the references to bring them into line with the journal's guidelines. None of the references comply with the MDPI format.
Finally, the comments made are intended to improve the quality of your work.

Author Response
Dear reviewer:
Thank you for the time you have dedicated to reviewing our manuscript. We have incorporated all the changes you suggested, which we believe have significantly improved the quality of our manuscript. Below, you will find our detailed responses to each of your comments, presented point by point. Additionally, all corresponding changes have been highlighted in yellow colour in the revised version of the manuscript.

Round 2
Reviewer 1 Report
Comments and Suggestions for Authors
Thanks for addressed every change proposed.
Author Response
We sincerely appreciate your thoughtful feedback and the time you devoted to reviewing our manuscript. We are delighted to know that our revisions have successfully addressed all your concerns. Your insightful comments were invaluable in improving the quality and clarity of our work.
Reviewer 2 Report
Comments and Suggestions for Authors
First of all, I would like to thank you for invited to read the document.
The authors have done an excellent job. The authors have been able to resolve all of my concerns and observations.
He considered that substantial modifications had been made to the document, which improved its methodological quality.
Note: It´s suggest reviewing reference 14 (Rugby), as it would be necessary to use references related to the subject of study.
It´s also suggested that the discussion and conclusions should mention that 24 sessions divided into 3 days/week x 8 weeks induce certain effects.
Author Response
Thank you sincerely for your kind words and the time you dedicated to reviewing our document. We are very pleased to hear that we were able to satisfactorily address all of your concerns. We greatly value your perspective, as your comments were essential in strengthening the manuscript.
